

# The influence of social power on neural responses to emotional conflict

Xueling Ma[1] and Entao Zhang[2,3]

[1] College of Psychology Liaoning Normal University, Dalian, China
[2] Henan University, Institute of Cognition, Brain and Health, Kaifeng, China
[3] Henan University, Institute of Psychology and Behavior, Kaifeng, China

## ABSTRACT

**Background:** Major power theories assume that social power can play an important role in an individual's goal-related behaviors. However, the specific psychological mechanisms through which this occurs remain unclear. Some studies suggested that having power enhanced individuals' goal-related behaviors, by contrast, other studies suggested that low-power individuals were associated with a greater performance in goal-directed tasks. We were particularly interested in how social power changes individuals' goal-related behaviors during an emotional face-word Stroop task.

**Method:** Social power was primed by asking participants to recall a past situation in which they were in a position of power (high-power individuals), or a situation in which they were lacking power (low-power individuals). Afterward, participants were asked to complete an emotional face-word Stroop task. In the task, words representing specific emotions were written in a prominent red color across a face, and these words and facial expressions were either congruent or incongruent. The participant's task was to judge the emotion of the face while ignoring the red emotional words.

**Results:** Our behavioral data showed that these individuals displayed faster reaction time and better accuracy in congruent conditions, slower reaction time for fearful faces and worse accuracy for happy faces in both incongruent and congruent conditions. The event-related potential analyses showed that, compared with low-power individuals, high-power individuals showed greater P1 amplitudes when faced with emotional stimuli (both incongruent and congruent conditions), indicating that power affects individuals' attention in the early sensory processing of emotional stimuli. For the N170 component, low-power individuals showed more negative amplitudes when facing emotional stimuli, indicated that low-power individuals paid more attention to the construct information of emotional stimuli. For the N450 component, compared with congruent conditions, incongruent conditions elicited more negative amplitudes for both high- and low-power individuals. More importantly, fearful faces provoked enhanced P1 amplitudes in incongruent conditions than in congruent conditions only for low-power individuals, while, happy faces elicited larger P1 amplitudes in congruent conditions than in incongruent conditions only for high-power individuals. The findings suggested that during the initial stage of stimuli processing low-power individuals are more sensitive to negative stimuli than high-power individuals.

**Conclusion:** These findings provided electrophysiological evidence that the differences in the emotional conflict process between high- and low-power individuals mainly lies in the early processing stages of emotional information.

Corresponding author
Entao Zhang,
10030094@vip.henu.edu.cn

Furthermore, evidence from P1 and N170 showed that there was also a redistribution of attentional resources in low-power individuals.

# INTRODUCTION

Social power is a basic social phenomenon characterized by influencing people and resources such as material (food, money, or economic opportunity) and social (knowledge, friendship, or decision-making opportunities) (*Keltner, Gruenfeld & Anderson, 2003*). Social power appears to facilitate individuals' ability to pursue goals (*Schmid, Kleiman & Amodio, 2015*; *Yin & Smith, 2019*).

The Situated Focus Theory of Power argues that power leads to situated behavior driven by the prioritization of salient goals and constructs (*Guinote, 2007a*). Social power promotes behavioral flexibility and individuals' behavior that is consistent with goals and motivations, their attention focus can vary with changes in different conditions (*Guinote, 2007a, 2008*; *Overbeck & Park, 2006*). In a previous study, they observed a higher behavior variability in high-power individuals than low-power individuals (*Guinote, Judd & Brauer, 2002*). *Guinote* found that, compared to low-power individuals, high-power individuals paid more attention to work (vs. social) information on a weekday and to social (vs. work) information on a leisure day. In contrast, low-power individuals tend to pay attention to more peripheral and detailed information, irrespective of its relevance (*Guinote, 2017*) as they are always under threat (*Guinote, 2017*). In line with this view, previous studies found that high-power individuals were associated with greater goal focus and performance in the presence of distractors on a variety of response conflict tasks (*Guinote, 2007a*; *Smith et al., 2008*; *Schmid, Kleiman & Amodio, 2015*).

In contrast, there is also increasing evidence suggesting that high-power individuals are less accurately in recognizing goal information (*Martin et al., 2012*; *Uskul, Paulmann & Weick, 2016*). For example, a recent study has shown that, in cognitive tasks where participants discriminated sets of objects based on shape, texture, and size, high-power participants performed worse than low-power participants (*Weick & Guinote, 2010*). More recently, *Uskul, Paulmann & Weick (2016)* provided evidence that high-power individuals are less accurately than low-power individuals when tested on recognition of emotional voices. Thus, the effects of power on information processing are complex and seem to vary in a task dependent manner (*Hall, Schmid & Latu, 2015*).

In the present study, we used the emotional-Stroop task to explore how power affects individuals' goal-related behavior. The emotional-Stroop task is often used by provoking interference through the semantic incompatibility between emotional faces and emotional words, and has been employed in many studies to investigate emotional conflict (*Hu et al., 2019*; *Stickel et al., 2019*; *Chechko et al., 2012, 2013*), the basic conclusions are that the reaction time for the incongruent condition is significantly higher than the congruent condition, while the accuracy for the incongruent condition is significantly

lower than the congruent condition (*Zhu et al., 2010*). This indicated that in the emotional Stroop task, emotional faces will affect individuals' attention. According to the relevant studies, the emotional-Stroop effect has been shown to activate certain brain mechanisms and to evoke multiple electroencephalography (EEG) components. For example, *Shen et al. (2013)* found that the N170 was mainly associated with conflicts. *Meeren, Van Heijnsbergen & De Gelder (2005)* found that when judging emotional faces, the body expression under the congruent condition can produced a larger P1 component in the early stage of perception than the incongruent body expression. Researchers believe that the N450 component may be related to an individual's conflict detection processing in the emotional conflict task (*Xue et al., 2015*).

Previous studies have examined the relationship between power and attention to others' emotions. However, we were interested in how high- and low-power affects individuals' attention to emotions in incongruent and congruent conditions. The current study took event-related potentials (ERPs) to provide accurate information about the time progress of the brain. According to relevant research results and the purpose of this study (*Shen et al., 2013*; *Meeren, Van Heijnsbergen & De Gelder, 2005*; *Xue et al., 2015*), the average amplitude of components of P1 (120–170 ms), N170 (150–180 ms) and N450 (350–450 ms) were analyzed. With reference to past studies (*Luo et al., 2010*; *Tillman & Wiens, 2011*), the P7 and P8 electrode points were selected for P1 and N170 analysis. The electrode points of F3, F4, Fz, C3, C4, Cz, CP1, and CP2 were selected for N450 analysis.

To summarize, the present study conducted an emotional conflict task to explore the attention bias in high- and low-power individuals. We hypothesized that (1) N170 amplitude in response to congruent conditions would be larger in the low-power individuals than in the high-power individuals at the P7 electrode; (2) P1 amplitude in response to congruent conditions would be larger in the low-power individuals than in the high-power individuals at the P8 electrode; (3) N450 amplitude in response to congruent conditions would not significantly different between low-power individuals and high-power individuals.

## METHOD

### Ethics statement

Data collection conformed to the Declaration of Helsinki and had been approved by the local Ethics Committee of Henan University (the approval number: HUSOM 2017-217), and all participants signed written informed consent before the experiment.

### Participants

G*Power 3.1 was used to estimate the sample size. For a 2 (social power: high vs. low) ×2 (emotional conflict: incongruent vs. congruent) two-factor mixed design, the significance level $\alpha = 0.05$, statistical test power 1- $b = 0.8$, a moderate effect size Cohen's $d = 0.25$, each group required 17 participants, the total sample size were 34. In order to improve the statistical test, we still have chosen 20 participants in each group.

A total of 40 college students participated in this experiment. Two participants' data were rejected due to intensive head movements during EEG recording. Finally,

38 participants' data were included (mean age = 21.4 years, SD = 1.23 years, 19 males) in this study, 19 each in the high-power (10 male) and low-power groups (9 male). All participants reported normal or corrected-to-normal vision, right-handed, and had no color vision deficiency or color blindness. Also, they reported no history of affective disorder and were free of any psychiatric medication. Each participant received 50 yuan after an hour of the experiment.

## Apparatus and stimuli

Facial images were selected from the Chinese Affective Picture System (CAPS), consisting of 16 "Happy" facial images (five females and five males) and 16 "Fear" facial images (five females and five males). They were cropped faces in a grayscale color. According to the literature (*Zhu et al., 2010*), the two Chinese words "愉快" (means "happy") and "恐惧" (means "fear") were written in red on each facial pictures. Finally, we got 64 facial pictures as experimental stimuli. The incongruent condition was when the facial expression was incongruent with the red Chinese characters; the reverse was the congruent condition. Among them, 32 images represented the congruent condition and 32 the incongruent condition. The emotional faces were placed in the center of a screen of 3.5° wide ×5° high, the Chinese words were approximately 1° (horizontal) ×1° (vertical). The size of face was 9.17 cm (horizontal) × 10.28 cm (vertical).

## Experimental procedures

Participants were randomly assigned to either a high-power individuals or low-power individuals. Following the power manipulation, participants performed the emotional-Stroop task and then completed a questionnaire assessing the power manipulation. Data were collected as previously described in *Ma, Wu & Zhang (2019)*, specifically in power manipulation and electroencephalography recordings.

## Power manipulation

Power was manipulated through a retrospective priming procedure (*Galinsky, Gruenfeld & Magee, 2003*; *Uskul, Paulmann & Weick, 2016*; *Hogeveen, Inzlicht & Obhi, 2014*). In a standardized way, the participants were asked to recall and describe a particular incident in which they had power over another individual (high power prime) or someone else had power over them (low power prime) during the previous week's events and, whether this experience was positive or negative. Emotional-Stroop task. The experimental procedure was controlled by E-Prime (Psychology Software Tools, Inc., Pittsburgh, PA, USA), this procedure consisted of 4 blocks, and every block consisted of 80 trials, with an equal proportion of each condition. Each trial began with the presentation of a fixation for 400 ms. After a random inter stimulus interval (ISI) between 400 and 600 ms, the stimulus was presented for 1,000 ms, and finally an inter stimulus interval (ISI) was presented for 1,800–2,300 ms. To avoid priming effects, direct repetitions of the same emotion-word-distractor combination were avoided. The participant's task was to rapidly and accurately judge the emotion of the face ("fear" or "happy"), while ignoring the red emotional words ("恐惧" or "愉快") written on the nose of the

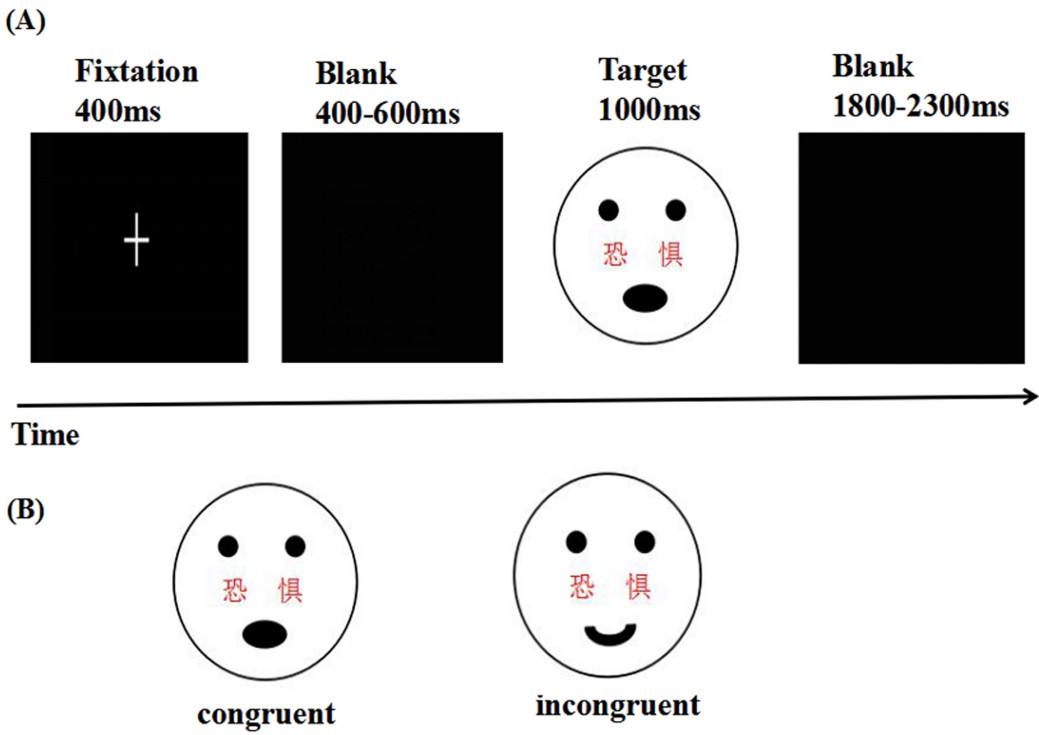

**Figure 1 Experimental procedures and the example stimuli.** (A) Sequence of events in a representative trial of the experiment, and (B) the example stimuli. Participants were asked to identify the faces with fearful or happy expressions that had either "fear" or "happy" written across them. Stimulus material consisted of congruent and incongruent conditions (In the actual experiment, the images were the real person pictures from the Chinese Affective Picture System).

faces, by pressing the "1" key with their right index finger (happy faces) or the "2" key with their right middle finger (fearful faces) (Fig. 1).

## Manipulation checks

After the Stroop task was completed, participants were asked to finish a questionnaire which contained two items ("Now I feel I have a great sense of power" and "Now I feel my wishes don't matter" (reverse scoring)) to complete the power manipulation check (*Kraus, Chen & Keltner, 2011*), responses were made using 7-point scales(from 1 = "strongly disagree" to 7 = "strongly agree") (α = 0.89). Finally, we used The Cognitive Emotion Regulation Questionnaire (CERQ) which contained 36-items to appraise participants' self-control ability (*Garnefski, Kraaij & Spinhoven, 2001*).

## Electroencephalography recordings and analysis

The electroencephalogram (EEG) was recorded from 32 scalp sites using electrodes mounted on an Ag/AgCl cap (Brain Product), EEG signal was recorded from electrodes arranged according to the standard 10–20 system. Electrodes were placed above and below on the right eye to record the vertical EOG (VEOG). All inter-electrode impedance was maintained below 5 kΩ. The EEG and EOG were amplified using a 0.01–30 Hz bandpass

and continuously sampled at 500 Hz. In the off-line analysis, the epoch of the analysis was from 200 ms pre-stimulus to 1,000 ms post-stimulus. In the experiment, the number of overlapping trials per participant under each stimulus condition was more than 60 times. The analysis was performed using social power (high vs. low) × emotional conflict (incongruent vs. congruent) × electrode three-factors repeated-measures ANOVA. In order to understand the results better, the data for the different conditions were analyzed separately: (1) all the faces: incongruent conditions vs. congruent conditions; (2) only fearful faces: incongruent conditions vs. congruent conditions; and (3) only happy faces: incongruent conditions vs. congruent conditions. *P* values were corrected by Greenhouse-Geisser correction. Post hoc analyses were conducted using Bonferroni corrected *t*-tests.

## RESULTS

### Manipulation checks

An ANOVA testing the power manipulation was significant, high-power individuals ($M = 4.74$, SD = 0.98) felt significantly more powerful than low-power individuals ($M = 4.03$, SD = 0.94), $t(36) = -2.29, p = 0.028$, $d = 0.74$, which indicated that the power manipulation was effective.

An ANOVA testing the effect of participants' self-control ability, there was no significant differences were found between high and low-power individuals ($t(36) = -0.99$, $p = 0.32$).

### Results of behavioral tests

We used PP graph and histogram to check normal distribution. The results suggested that our data basically conforms to the normal distribution. A mixed-model ANOVA with average reaction time as the dependent variable, power as the between-subjects factor, and emotional conflict and emotion as within-subject factors. The main effect of power was not significant, $F(1, 36) = 0.818$, $p > 0.05$, $\eta2 = 0.022$. There was a significant main effect of emotional conflict ($F(1, 36) = 54.238$. $p < 0.001$, $\eta2 = 0.601$), reflecting that reaction times in the congruent condition($M = 667$, SE = 8) were significantly faster than in the incongruent condition($M = 697$, SE = 9). There was also a significant effect of emotion ($F(1, 36) = 4.473$, $p = 0.04$, $\eta2 = 0.11$), showed that the fearful faces ($M = 691$, SE = 9) processed more slowly than happy faces ($M = 673$, SE = 10).

For accuracy rate analyses, the three-way mixed-model ANOVA revealed that the main effect of power was not significant, $F(1, 36) = 0.005$, $p > 0.05$, $\eta2 = 0.000$. In addition, we found a significant effect of emotional conflict ($F(1, 36) = 46.807$, $p < 0.001$, $\eta2 = 0.565$) and of emotion ($F(1, 36) = 9.222$, $p = 0.004$, $\eta2 = 0.204$), indicating that a higher level of correct responses during congruent (91.2%) compared to incongruent (82.1%) conditions and greater accuracy for fear (88.4%) compared to happy(84.9%) faces. The interaction effect of emotional conflict × emotion was significant, $F(1, 36) = 5.507$, $p = 0.025$, $\eta2 = 0.133$. Bonferroni-corrected pairwise comparison indicated a higher level of correct responses during congruent (92.3%) compared to the incongruent conditions

(84.6%) ($p < 0.001$) in fearful faces and greater accuracy for congruent (90.1%) compared to the incongruent conditions (79.7%) in happy faces ($p < 0.001$) (Fig. 2). There were no other significant results between the RTs and ACC.

## ERP results

(1) All the faces: incongruent conditions vs. congruent conditions

For the P1 component, the main effect of power was significant ($F$ (1, 36) = 2.937, $p = 0.095$, η2 = 0.075), the emotional stimuli for high-power individuals primed ($M = 0.137$, SE = 0.693) a more positive amplitude than for low-power individuals ($M = -1.544$, SE = 0.693). The main effect of emotional conflict ($F$ (1, 36) = 0.042, $p = 0.838$), the interaction of power × emotional conflict ($F$ (1, 36) = 1.884, $p = 0.178$), the effect of electrode ($F$ (1, 36) = 0.087, $p = 0.769$) did not reach significance.

For the N170 component, the main effect of power was significant, $F$ (1, 36) = 3.291, $p = 0.078$, η2 = 0.084, the emotional stimuli for low-power individuals elicited a more negative N170 ($M = -5.466$, SE = 0.992) than for high-power individuals ($M = -2.922$, SE = 0.992). Main effect of emotional conflict ($F$ (1, 36) = 0.191, $p = 0.665$), and the interaction of power × emotional conflict ($F$ (1, 36) = 0.386, $p = 0.538$) were not significant. Meanwhile, a significant main effect of electrode was observed, $F$(1, 36) = 3.164, $p = 0.084$, η2 = 0.081, the largest amplitudes were elicited at the P8 (−4.757 μV) electrode site.

For the N450 component, the main effect of power ($F$ (1, 36) = 0.047, $p = 0.829$) were not significant, while the main effect of emotional conflict was significant, $F$ (1, 36) = 5.915, $p = 0.02$, η2 = 0.141, the amplitude was enhanced in the incongruent condition ($M = 3.262$, SE = 0.484) than in the congruent condition ($M = 3.806$, SE = 0.468). The interaction of power × emotional conflict ($F$ (1, 36) = 0.79, $p = 0.38$) did not reach significance. Meanwhile, a significant main effect of electrode was observed, $F$ (7, 252) = 29.809, $p = 0.000$, η2 = 0.453, suggesting that the largest N450 amplitudes on CP2 ($M = 6.56$ μV, SE = 0.55) (Figs. 3 and 4).

(2) Only fearful faces: incongruent conditions vs. congruent conditions

For the P1 component, the main effect of power ($F$ (1, 36) = 2.469, $p = 0.125$) were not significant, while the main effect of emotional conflict was significant, $F$ (1, 36) = 6.557, $p = 0.015$, η2 = 0.154, the amplitude was enhanced in the congruent condition ($M = -0.842$, $SE = 0.489$) than in the incongruent condition ($M = -0.573$, SE = 0.512). The effect of electrode did not reach significance, $F$ (1, 36) = 0.185, $p = 0.67$. The interaction of power × emotional conflict was significant, $F$ (1, 36) = 4.134, $p = 0.049$, η2 = 0.103. Specific contrasts revealed that the P1 amplitude was significantly larger in incongruent conditions than in congruent conditions only for low-power individuals ($F$(1, 36) = 10.552, $p = 0.003$, η2 = 0.227; incongruent conditions: $M = -1.249$ μV, SE = 0.724; congruent conditions: $M = -1.731$ μV, SE = 0.692) but not for high-power individuals ($F$(1, 36) = 0.139, $p = 0.711$, η2 = 0.004; incongruent conditions: $M = 0.102$ μV, SE = 0.724; congruent conditions: $M = 0.047$ μV, SE = 0.692).
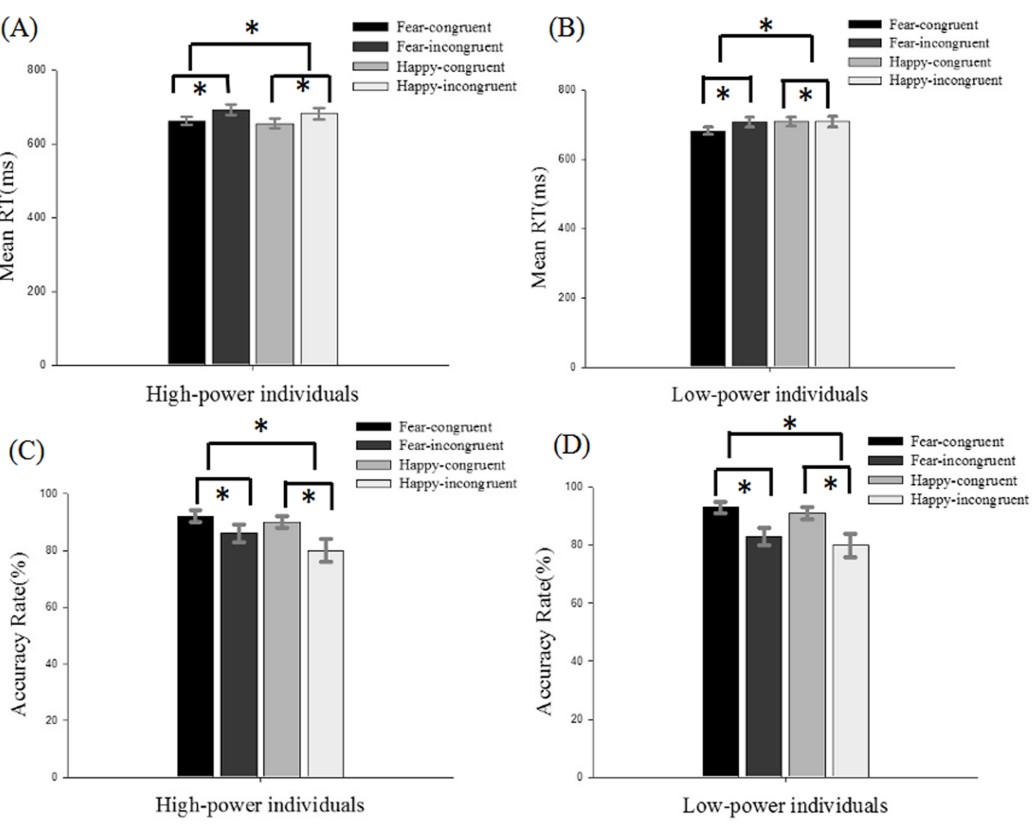

**Figure 2 Descriptive statistics for response times (ms) and accuracy rate (%) for high- and low-power individuals.** (A) Descriptive statistics for response times (ms) for high-power individuals and (B) for low-power individuals, and (C) accuracy rate (%) for high-power individuals and (D) for low-power individuals. *: $p < 0.05$.

For the N170 component, the main effect of power was significant, $F(1, 36) = 3.394$, $p = 0.074$, $\eta2 = 0.086$, the emotional stimuli for low-power individuals elicited a more negative N170 ($M = -5.45$, SE = 1.012) than for high-power individuals ($M = -2.813$, SE = 1.012). Main effect of emotional conflict was significant, $F(1, 36) = 4.187$, $p = 0.048$, $\eta2 = 0.104$, congruent conditions elicited a more negative N170 ($M = -4.27$, SE = 0.705) than incongruent conditions ($M = -3.992$, SE = 0.733). The interaction of power × emotional conflict ($F(1, 36) = 1.447$, $p = 0.237$) was not significant. Meanwhile, the effect of electrode was not significant, $F(1, 36) = 2.024$, $p = 0.163$.

For the N450 component, the main effect of power ($F(1, 36) = 0.166$, $p = 0.686$) were not significant, while the main effect of emotional conflict was significant, $F(1, 36) = 7.165$, $p = 0.011$, $\eta2 = 0.166$, the amplitude was enhanced in the incongruent condition ($M = 3.058$, SE = 0.49) than in the congruent condition ($M = 3.76$, SE = 0.501). The interaction of power × emotional conflict ($F(1, 36) = 0.816$, $p = 0.372$) was not significant. Meanwhile, the effect of electrode was significant, $F(7, 252) = 29.585$, $p = 0.000$, $\eta2 = 0.45$, suggesting that the largest N450 amplitudes on CP2 ($M = 6.479$ μV, SE = 0.559) (Figs. 5 and 6).

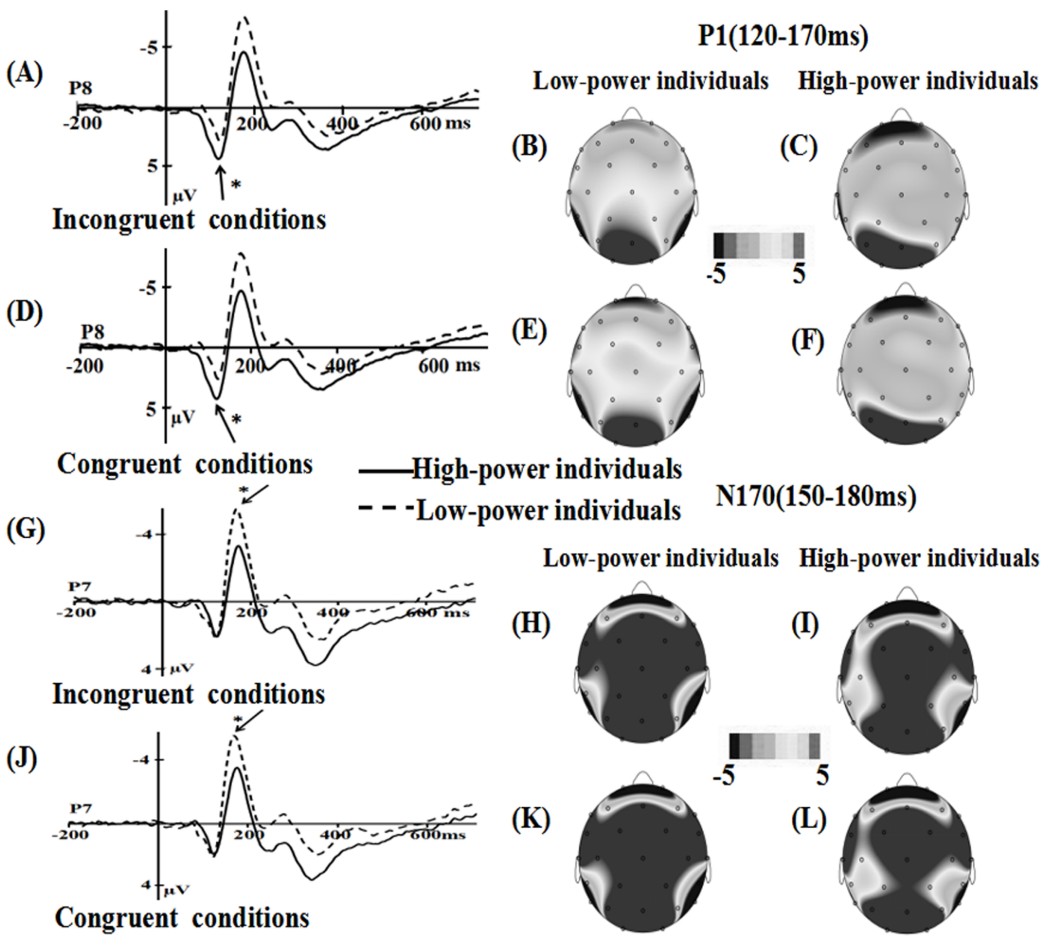

**Figure 3  Waveforms and topographical maps of P1 and N170 components for all the faces.** (A) The Grand-average P1 in incongruent conditions at P8 electrode site for all the faces, and (B) Topographical maps of the P1 for low-power individuals and (C) for high-power individuals in incongruent conditions. (D) The Grand-average P1 in congruent conditions at P8 electrode site for all the faces, and (E) topographical maps of the P1 for low-power individuals and (F) for high-power individuals in congruent conditions. (G) The Grand-average N170 in incongruent conditions at P7 electrode site for all the faces, and (H) Topographical maps of the N170 for low-power individuals and (I) for high-power individuals in incongruent conditions. (J) The Grand-average N170 in congruent conditions at P7 electrode site for all the faces, and (K) topographical maps of the N170 for low-power individuals and (L) for high-power individuals in congruent conditions. *: $p < 0.05$.

(3) Only happy faces: incongruent conditions vs. congruent conditions.

For the P1 component, the main effect of power was significant ($F_{(1, 36)} = 3.381$, $p = 0.074$, $\eta2 = 0.086$), the emotional stimuli for high-power individuals primed ($M = 0.201$, $SE = 0.692$) a more positive amplitude than for low-power individual ($M = -1.599$, $SE = 0.692$). The main effect of emotional conflict ($F_{(1, 36)} = 2.784$, $p = 0.104$), the interaction of power × emotional conflict ($F_{(1, 36)} = 0.024$, $p = 0.878$), the effect of electrode ($F_{(1, 36)} = 0.024$, $p = 0.878$) did not reach significance.

The interaction of power × emotional conflict × electrode was significant, ($F_{(1, 36)} = 5.70$, $p = 0.022$, $\eta2 = 0.137$). Specific contrasts revealed that the P1 amplitude was

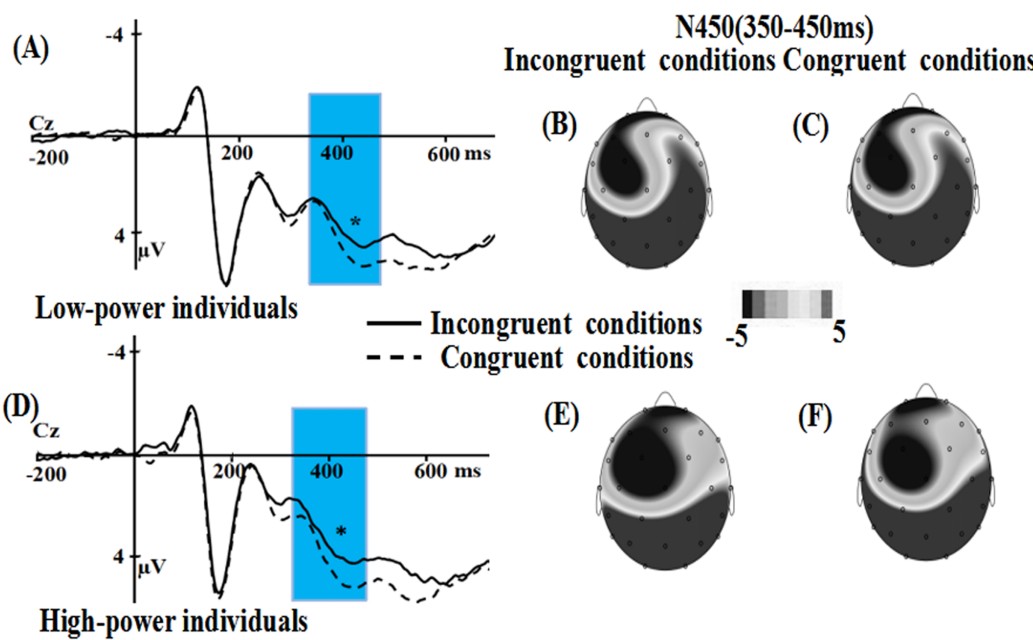

**Figure 4 Waveforms and topographical maps of N450 components for all the faces.** (A) The Grand-average N450 in low-power individuals at CZ electrode site for all the faces, and (B)Topographical maps of the N450 in incongruent conditions and (C) in congruent conditions for low-power individuals. (D) The Grand-average N450 in high-power individuals at CZ electrode site for all the faces, and (E) topographical maps of the N450 in incongruent conditions and (F) in congruent conditions for high-power individuals. *: $p < 0.05$.

significantly larger in congruent conditions than in incongruent conditions only for high-power individuals ($F(1, 36) = 5.203$, $p = 0.029$, η2 = 0.126; incongruent conditions: $M = 0.111$ μV, SE = 0.914; congruent conditions: $M = 0.629$ μV, SE = 0.944) but not for low-power individuals ($F(1, 36) = 0.12$, $p = 0.731$, η2 = 0.003; incongruent conditions: $M = −1.728$ μV, SE = 0.914; congruent conditions: $M = −1.65$ μV, SE = 0.944) at P8 electrode.

For the N170 component, the main effect of power was significant, $F(1, 36) = 3.149$, $p = 0.084$, η2 = 0.08, the emotional stimuli for low-power individuals showed a more negative N70 ($M = −5.483$, SE = 0.978) than high-power individuals ($M = −3.029$, SE = 0.978) did. Main effect of emotional conflict ($F (1, 36) = 1.613$, $p = 0.212$), and the interaction of power × emotional conflict ($F (1, 36) = 0.058$, $p = 0.811$) were not significant. Meanwhile, a significant main effect of electrode was observed, $F(1, 36) = 4.294$, $p = 0.045$, η2= 0.107, the largest amplitudes were elicited at the P8 (−4.949 μV) electrode site.

For the N450 component, the main effect of power ($F (1, 36) = 0.000$, $p = 0.986$), the main effect of emotional conflict ($F (1, 36) = 2.307$, $p = 0.138$), the interaction of power × emotional conflict ($F (1, 36) = 0.411$, $p = 0.526$) did not reach significance. Meanwhile, a significant main effect of electrode was observed, $F(7, 252) = 28.243$, $p = 0.000$, η2 = 0.44, the largest amplitudes were elicited on CP2 ($M = 6.64$ μV, SE = 0.555) electrode site (Fig. 7).

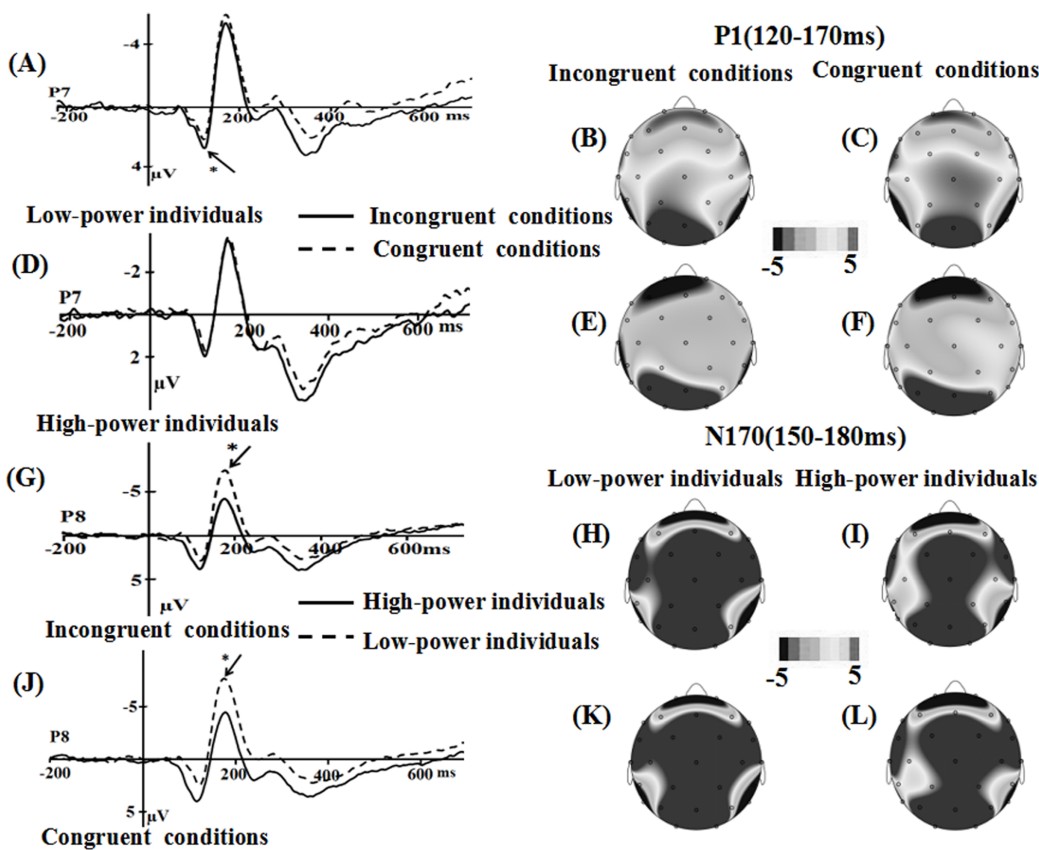

**Figure 5 Waveforms and topographical maps of P1 and N170 components for fear faces.** (A) The Grand-average P1 in low-power individuals at P7 electrode site for fear faces, and (B) topographical maps of the P1 in incongruent conditions and (C) in congruent conditions for low-power individuals. (D) The Grand-average P1 in high-power individuals at P7 electrode site for fear faces, and (E) topographical maps of the P1 in incongruent conditions and (F) in congruent conditions for high-power individuals. (G) The Grand-average N170 in incongruent conditions at P8 electrode site for fear faces, and (H) topographical maps of the N170 for low-power individuals and (I) for high-power individuals in incongruent conditions. (J) The Grand-average N170 in congruent conditions at P8 electrode site for fear faces, and (K) topographical maps of the N170 for low-power individuals and (L) for high-power individuals in congruent conditions. *: $p < 0.05$.     

To evaluate the strength of the empirical evidence, we also conducted a median analysis and a non-parametric test (Mann–Whitney Test). Specifically, the median analysis and the Mann–Whitney test suggested that for RT, there was a significant difference in low- and high-power individuals only in incongruent conditions when individuals faced happy faces (see Tables 1 and 2). Considering the ERP results, the results supported the difference between low-power and high-power individuals when faced fearful faces on N170 and in all conditions on P1 components (see Tables 1 and 2).

## DISCUSSION

The goal of this study was to test how power facilitates individuals' attention in congruent and incongruent conditions. In the emotional Stroop task, the participants, reacted more slowly and less accurately to incongruent conditions compared to congruent

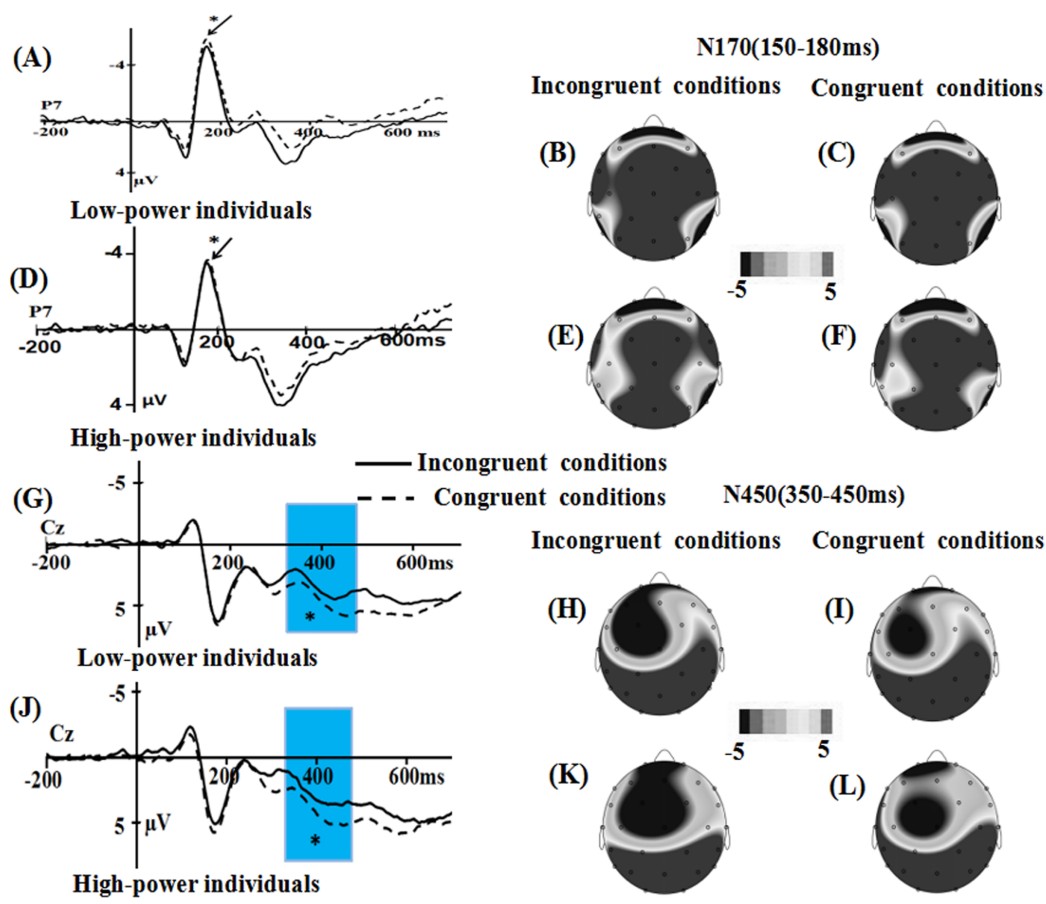

**Figure 6 Waveforms and topographical maps of N170 and N450 components for fear faces.** (A) The Grand-average N170 in low-power individuals at P7 electrode site for fear faces, and (B) topographical maps of the N170 in incongruent conditions and (C) in congruent conditions for low-power individuals. (D) The Grand-average N170 in high-power individuals at P7 electrode site for fear faces, and (E) topographical maps of the N170 in incongruent conditions and (F) in congruent conditions for high-power individuals. (G) The Grand-average N450 in low-power individuals at CZ electrode site for fear faces, and (H) topographical maps of the N450 in incongruent conditions and (I) in congruent conditions for low-power individuals. (J) The Grand-average N450 in high-power individuals at CZ electrode site for fear faces, and (K) topographical maps of the N450 in incongruent conditions and (L) in congruent conditions for high-power individuals. *: $p < 0.05$.

conditions. Therefore, the present research is certainly consistent with the existing results (*Shen et al., 2013*; *Zhou et al., 2015*). There were, however, fewer errors overall in the fear face compared to the happy face, suggesting that when it came to targets, attention was likely facilitated by the negative emotional features of the faces based on which participants were required to make their judgment.

However, electrophysiological evidence showed the distinction between high-power and low-power individuals. Specifically, smaller P1 amplitudes were observed for low-power individuals than high-power individuals. The P1 component is considered to reflect the early phase of sensory and perceptual processing of emotional stimuli (*Luo et al., 2010*; *Rellecke, Sommer & Schacht, 2012*). Thus, we inferred that power affected individuals' attention during the sensory processing in both incongruent and congruent conditions.

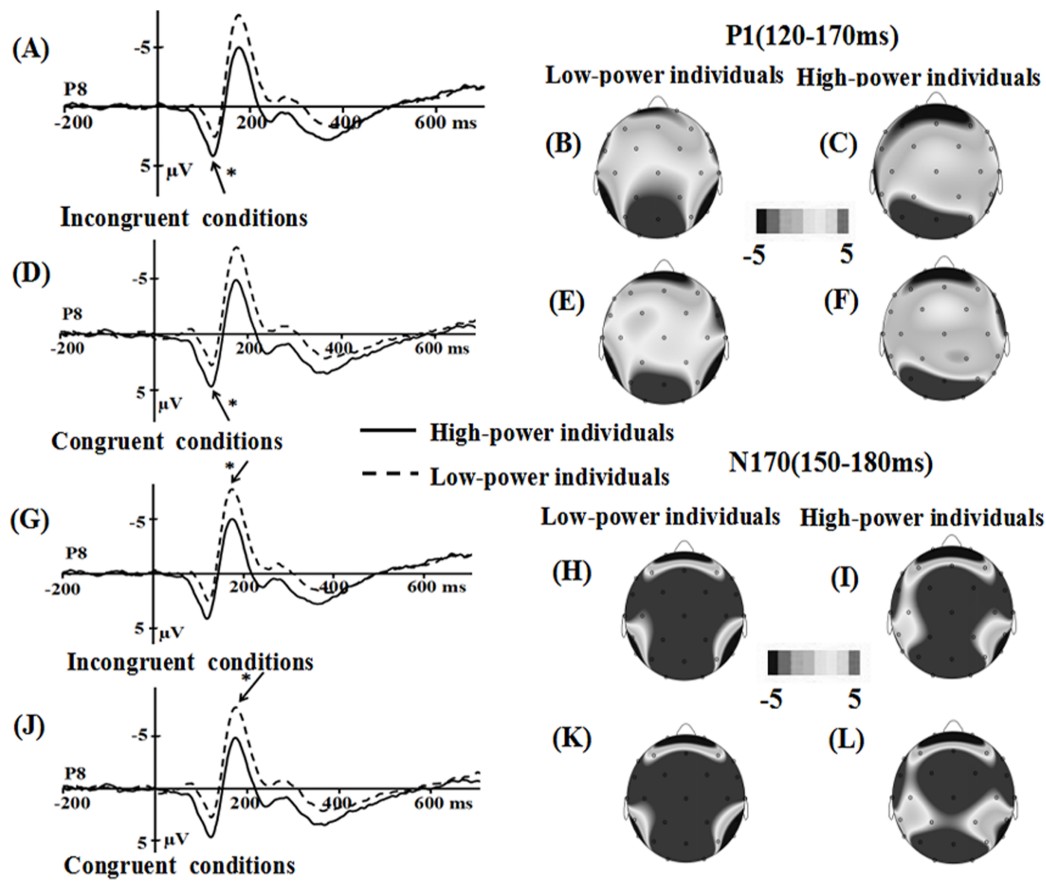

**Figure 7 Waveforms and topographical maps of P1 and N170 components for happy faces.** (A) The Grand-average P1 in incongruent conditions at P8 electrode site for happy faces, and (B) topographical maps of the P1 for low-power individuals and (C) for high-power individuals in incongruent conditions. (D) The Grand-average P1 in congruent conditions at P8 electrode site for happy faces, and (E) topographical maps of the P1 for low-power individuals and (F) for high-power individuals in congruent conditions. (G) The Grand-average N170 in incongruent conditions at P8 electrode site for happy faces, and (H) topographical maps of the N170 for low-power individuals and (I) for high-power individuals in incongruent conditions. (J) The Grand-average N170 in congruent conditions at P8 electrode site for happy faces, and (K) topographical maps of the N170 for low-power individuals and (L) for high-power individuals in congruent conditions. *: $p < 0.05$.

The experimental results could be suggested that more attentional resources were allocated in the early stage in high-power individuals. This result was consistent with the situated focus theory of power and previous findings, which high-power individuals could alter their attentional focus as a function of the demands of the situation. Regarding the N170 component, the low-power individuals showed a more negative N170 amplitude in both congruent and incongruent conditions, indicating that low-power individuals were more concerned with identifying the features of the emotional stimulus itself than high-power individuals, which attenuated the flexibility of attention. Generally, in the emotional conflict Stroop task the N450 component was related to response conflict processing (*Chen et al., 2011*). More specifically, N450 component was reported to be associated with response conflict detection and resolution (*Coderre, Conklin & Van*

**Table 1 The median for all conditions.**

| Power | Emotional conflict | The median (ms) | |
|---|---|---|---|
| | | Fear faces | Happy faces |
| All the faces in RT | | | |
| High power | Congruent | 680 | 653 |
| | Incongruent | 704 | 681 |
| Low power | Congruent | 696 | 678 |
| | Incongruent | 737 | 736 |

| Power | Emotional conflict | The median (%) | |
|---|---|---|---|
| | | Fear faces | Happy faces |
| All the faces in ACC | | | |
| High power | Congruent | 94 | 95 |
| | Incongruent | 93 | 88 |
| Low power | Congruent | 94 | 91 |
| | Incongruent | 85 | 81 |

| Power | Emotional conflict | The median (μV) | |
|---|---|---|---|
| | | P7 | P8 |
| All the faces in P1 | | | |
| High power | Congruent | 0.36 | −0.23 |
| | Incongruent | 0.32 | 0.13 |
| Low power | Congruent | −1.8 | −0.87 |
| | Incongruent | −1.26 | −1.04 |

| Power | Emotional conflict | The median (μV) | |
|---|---|---|---|
| | | P7 | P8 |
| All the faces in N170 | | | |
| High power | Congruent | −1.41 | −3.57 |
| | Incongruent | −1.71 | −3.98 |
| Low power | Congruent | −3.63 | −5.52 |
| | Incongruent | −4.64 | −6.56 |

| Power | Emotional conflict | The median (μV) | | | | | | | |
|---|---|---|---|---|---|---|---|---|---|
| | | F3 | F4 | FZ | C3 | C4 | CZ | CP1 | CP2 |
| All the faces in N450 | | | | | | | | | |
| High power | Congruent | 1.38 | 1.59 | 0.92 | 2.99 | 5.16 | 5.50 | 6.54 | 7.12 |
| | Incongruent | 0.38 | 1.03 | 0.40 | 1.81 | 3.39 | 4.12 | 5.35 | 6.50 |
| Low power | Congruent | −0.67 | 2.27 | 0.70 | 2.70 | 3.94 | 3.71 | 5.60 | 6.45 |
| | Incongruent | −1.03 | 1.54 | −0.91 | 2.09 | 3.56 | 3.27 | 5.35 | 6.02 |

| Power | Emotional conflict | The median (μV) | |
|---|---|---|---|
| | | P7 | P8 |
| Only fearful faces in P1 | | | |
| High power | Congruent | 0.12 | 0.58 |
| | Incongruent | 0.56 | 0.36 |

| Power | Emotional conflict | The median (μV) | |
| --- | --- | --- | --- |
| | | **P7** | **P8** |
| Low power | Congruent | −1.66 | −0.90 |
| | Incongruent | 0.86 | −1.13 |

| Power | Emotional conflict | The median (μV) | |
| --- | --- | --- | --- |
| | | **P7** | **P8** |
| **Only fearful faces in N170** | | | |
| High power | Congruent | −1.84 | −2.89 |
| | Incongruent | −1.71 | −3.19 |
| Low power | Congruent | −3.84 | −5.35 |
| | Incongruent | −4.20 | −5.72 |

| Power | Emotional conflict | The median (μV) | | | | | | | |
| --- | --- | --- | --- | --- | --- | --- | --- | --- | --- |
| | | **F3** | **F4** | **FZ** | **C3** | **C4** | **CZ** | **CP1** | **CP2** |
| **Only fearful faces in N450** | | | | | | | | | |
| High power | Congruent | 1.19 | 1.47 | 1.04 | 2.94 | 4.78 | 5.66 | 6.21 | 7.09 |
| | Incongruent | 0.63 | 1.09 | 0.40 | 1.97 | 3.51 | 3.96 | 5.19 | 6.09 |
| Low power | Congruent | 0.72 | 2.01 | 0.20 | 2.85 | 4.83 | 4.39 | 5.55 | 7.08 |
| | Incongruent | −1.16 | 2.44 | −0.10 | 2.30 | 4.58 | 3.00 | 5.47 | 6.06 |

| Power | Emotional conflict | The median (μV) | |
| --- | --- | --- | --- |
| | | **P7** | **P8** |
| **Only happy faces in P1** | | | |
| High power | Congruent | 0.27 | −0.05 |
| | Incongruent | 0.44 | 0.13 |
| Low power | Congruent | −1.68 | −1.10 |
| | Incongruent | −1.35 | −1.36 |

| Power | Emotional conflict | The median (μV) | |
| --- | --- | --- | --- |
| | | **P7** | **P8** |
| **Only happy faces in N170** | | | |
| High power | Congruent | −1.54 | −4.25 |
| | Incongruent | −1.67 | −4.34 |
| Low power | Congruent | −3.41 | −5.69 |
| | Incongruent | −4.67 | −7.15 |

| Power | Emotional conflict | The median (μV) | | | | | | | |
| --- | --- | --- | --- | --- | --- | --- | --- | --- | --- |
| | | **F3** | **F4** | **FZ** | **C3** | **C4** | **CZ** | **CP1** | **CP2** |
| **Only happy faces in N450** | | | | | | | | | |
| High power | Congruent | 1.39 | 1.08 | 1.30 | 3.36 | 4.98 | 4.22 | 6.36 | 7.15 |
| | Incongruent | 0.89 | 1.61 | 1.11 | 2.35 | 3.85 | 4.28 | 5.52 | 6.78 |
| Low power | Congruent | −0.81 | 1.02 | 0.24 | 2.55 | 3.86 | 3.25 | 5.60 | 6.29 |
| | Incongruent | −0.83 | 0.67 | −0.01 | 2.17 | 3.18 | 3.46 | 5.65 | 6.26 |

**Table 2 Non-parametric test for all conditions.**

| Power | Emotional conflict | Z | P |
|---|---|---|---|
| **All the faces in RT** | | | |
| Fear faces | Congruent | −0.63 | 0.53 |
| | Incongruent | −0.95 | 0.34 |
| Happy faces | Congruent | −1.12 | 0.26 |
| | Incongruent | −1.74 | 0.08 |
| **All the faces in ACC** | | | |
| Fear faces | Congruent | −0.04 | 0.97 |
| | Incongruent | −0.99 | 0.32 |
| Happy faces | Congruent | −0.54 | 0.59 |
| | Incongruent | −0.34 | 0.74 |
| **All the faces in P1** | | | |
| P7 | Congruent | −1.42 | 0.157 |
| | Incongruent | −1.15 | 0.249 |
| P8 | Congruent | −1.80 | 0.073 |
| | Incongruent | −1.77 | 0.077 |
| **All the faces in N170** | | | |
| P7 | Congruent | −1.24 | 0.215 |
| | Incongruent | −1.30 | 0.194 |
| P8 | Congruent | −1.59 | 0.112 |
| | Incongruent | −1.59 | 0.112 |
| **All the faces in N450** | | | |
| F3 | Congruent | −0.95 | 0.343 |
| | Incongruent | −0.66 | 0.511 |
| F4 | Congruent | −2.20 | 0.827 |
| | Incongruent | −0.19 | 0.849 |
| FZ | Congruent | −0.48 | 0.630 |
| | Incongruent | −0.42 | 0.672 |
| C3 | Congruent | −0.25 | 0.804 |
| | Incongruent | −0.34 | 0.737 |
| C4 | Congruent | −0.25 | 0.804 |
| | Incongruent | −0.31 | 0.759 |
| CZ | Congruent | −0.10 | 0.919 |
| | Incongruent | −0.13 | 0.895 |
| CP1 | Congruent | −0.63 | 0.530 |
| | Incongruent | −1.10 | 0.919 |
| CP2 | Congruent | −0.02 | 0.988 |
| | Incongruent | −0.25 | 0.804 |
| **Only fearful faces in P1** | | | |
| P7 | Congruent | −1.42 | 0.157 |
| | Incongruent | −0.83 | 0.405 |
| P8 | Congruent | −2.03 | 0.042 |

| Table 2 (continued) | | | |
|---|---|---|---|
| **Power** | **Emotional conflict** | **Z** | **P** |
| | Incongruent | −1.62 | 0.11 |
| Only fearful faces in N170 | | | |
| P7 | congruent | −1.56 | 0.118 |
| | incongruent | −1.01 | 0.314 |
| P8 | congruent | −1.74 | 0.082 |
| | incongruent | −1.68 | 0.093 |
| Only fearful faces in N450 | | | |
| F3 | Congruent | −0.45 | 0.651 |
| | Incongruent | −0.37 | 0.715 |
| F4 | Congruent | −0.39 | 0.693 |
| | Incongruent | −0.48 | 0.630 |
| FZ | Congruent | −0.04 | 0.965 |
| | Incongruent | −0.04 | 0.965 |
| C3 | Congruent | −0.16 | 0.872 |
| | Incongruent | −0.13 | 0.895 |
| C4 | Congruent | −0.16 | 0.872 |
| | Incongruent | −0.63 | 0.530 |
| CZ | Congruent | −0.19 | 0.849 |
| | Incongruent | −0.31 | 0.759 |
| CP1 | Congruent | −0.51 | 0.609 |
| | Incongruent | −0.19 | 0.849 |
| CP2 | Congruent | −0.02 | 0.988 |
| | Incongruent | −0.25 | 0.804 |
| Only happy faces in P1 | | | |
| P7 | Congruent | −1.50 | 0.133 |
| | Incongruent | −1.77 | 0.077 |
| P8 | Congruent | −1.68 | 0.093 |
| | Incongruent | −1.65 | 0.099 |
| Only happy faces in N170 | | | |
| P7 | Congruent | −0.95 | 0.343 |
| | Incongruent | −1.45 | 0.140 |
| P8 | Congruent | −1.33 | 0.184 |
| | Incongruent | −1.47 | 0.140 |
| Only happy faces in N450 | | | |
| F3 | Congruent | −1.53 | 0.125 |
| | Incongruent | −0.92 | 0.358 |
| F4 | Congruent | −0.02 | 0.988 |
| | Incongruent | −0.10 | 0.919 |
| FZ | Congruent | −0.57 | 0.569 |
| | Incongruent | −0.66 | 0.511 |
| C3 | Congruent | −0.45 | 0.651 |

(Continued)

| Table 2 (continued) | | | |
|---|---|---|---|
| Power | Emotional conflict | Z | P |
| | Incongruent | −0.10 | 0.919 |
| C4 | Congruent | −0.07 | 0.942 |
| | Incongruent | −0.04 | 0.965 |
| CZ | Congruent | −0.31 | 0.759 |
| | Incongruent | −0.10 | 0.919 |
| CP1 | Congruent | −0.89 | 0.373 |
| | Incongruent | −2.77 | 0.782 |
| CP2 | Congruent | −0.39 | 0.693 |
| | Incongruent | −0.16 | 0.872 |

Heuven, 2011). Consistent with the previous ERP studies, the present study found that amplitudes in the N450 components in response to the incongruent condition were larger than to the congruent condition for all of the individuals (Xue et al., 2013, 2015). A possible reason could be that during the early stages, the high-power individuals were required to devote greater attentional resources in the emotional conflict task. However, as time passed and the participants entered the late component stage, the emotional conflict was gradually resolved with the continuous increase in attentional resources (Zhou et al., 2015). This indicated that attentional resource is one of the major factors influencing emotional conflict. Hence, the attentional resources accumulated during the P1 component in the high-power individuals had a facilitating effect on conflict processing.

In addition, the most important result of this study was that in fearful faces, low-power individuals tended to show larger amplitudes than high-power individuals did, and in happy faces, high-power individuals showed larger amplitudes than low-power individuals in the earlier P1 stage. Some ERP studies have shown evidence for an enhanced P1 component for negative relative to neutral stimuli (Vuilleumier, 2005). Our finding suggested that in the initial stage of negative stimuli processing low-power individuals are more sensitive than high-power individuals. Thus, together with work by Guinote (2017), our study at least demonstrated that social power affected individuals' sensory stage.

There were several limitations requiring discussion. On the one hand, the processing of emotional expressions is not only affected by individual power, but also biased by target power (emotion expresser) (Ratcliff et al., 2012a, 2012b). The situated focus theory of power proposed that high-power individuals are supposed to dynamically adapt in a complex, goal-congruent way to the target's power (Côté et al., 2011; Guinote, 2010). On the other hand, without the neutral (power control) condition, there is reason to doubt that the modulated effect of power on emotional conflict tasks might also be found in the power control condition, which would make the results less convinced.

## CONCLUSION

In summary, utilizing high temporal-resolution ERP technology, we provided solid evidence to support the view that social power can be an important factor in affecting

emotional conflict in the early processing stages of emotional information. Our results suggested that there was a redistribution of attentional resources in low power individuals. The findings above have further enriched the theoretical research on the relationship between social power and attention. Based on these results, we proposed that power effects on individuals' attention are an important goal for future research. Our findings could help to revise and qualify existing theories of how power affects attention.

### Funding

This work was supported by the National Natural Science Foundation of China (31700952) and the Philosophy and Social Science Foundation of Henan Province, China (2018BJY008). The funders had no role in study design, data collection and analysis, decision to publish, or preparation of the manuscript.

### Grant Disclosures

The following grant information was disclosed by the authors:
National Natural Science Foundation of China: 31700952.
Philosophy and Social Science Foundation of Henan Province, China: 2018BJY008.

### Competing Interests

The authors declare that they have no competing interests.

### Author Contributions

- Xueling Ma conceived and designed the experiments, performed the experiments, analyzed the data, prepared figures and/or tables, authored or reviewed drafts of the paper, and approved the final draft.
- Entao Zhang conceived and designed the experiments, authored or reviewed drafts of the paper, and approved the final draft.

### Human Ethics

The following information was supplied relating to ethical approvals (i.e., approving body and any reference numbers):

Data collection conformed to the Declaration of Helsinki and had been approved by the local Ethics Committee of Henan University (HUSOM 2017-217).

### Data Availability

The raw measurements are available in the Supplemental File.

### Supplemental Information

Supplemental information for this article can be found online at http://dx.doi.org/10.7717/peerj.11267#supplemental-information.

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
