# Peer review of "The influence of social power on neural responses to emotional conflict"

_PeerJ, doi:10.7717/peerj.11267_

## Round 0.1 · original submission · Major Revisions

Dear Dr. Zhang,

Thank you for your submission to PeerJ.

We have received two review comments and the reviewers are mostly concerned about the analyses and methods described. In my opinion, also the Article requires a Major revision keeping the comments of the reviewers.

Thank you,
Kind regards,
Kenichi Shibuya, PhD

·

Basic reporting

see below

Experimental design

see below

Validity of the findings

see below

Additional comments

Dear Xueling and Entao

Here are my comments as to your submission “the influence of social power on neural responses to emotional conflict”:
• The English level of the entire manuscripts needs professional editing. As it stands now, there are many English grammar errors and spelling (e.g. ‘colloege’ instead of ‘college’ and lines 363-364; but there are more).
• Lines 135-139: add to each of the 2 hypothesis that the effect is expected across all channels for each of the three waveforms. So for example ‘(1)’ would read ‘N170 amplitude in response to congruent emotion expressions would be 136 larger in low power group than in high one across the two channels’
• Power analyses: the design for the power was a 2 x 2; but the ERP analyses used a 2 x 2 x C; where ‘C’ stands for the number of channels used in each of the three waveforms. It seems the power analysis was designed thinking of RTs and accuracy rates as the response. But given than ‘channel’ is an extra factor for the microvoltage data, then another power analysis is needed. So for example, for the cases of N170 and P1 the design needed is 2 x 2 x 2 (as there are 2 channels) whereas for N450, it’d be 2 x 2 x 8 (as there were 8 channels). These new power analyses will dictate a new number of participants needed and will therefore impact the ERP analyses.
• Line 154: “There were nineteen participants in each group: high-power, low-power” what percentage of males in each?
• Line 156: “All participants were right-handed, did not have color vision deficiency or color blindness and had normal or corrected to normal vision” how was this determined? Self-report?
• Line 158: “In the end study, all participants were paid for this time.” How long did the study go for? And how much was each participant paid?
• Somewhere in the ‘methods’ state that SPSS was used for the analyses. On this matter, please also upload the SPSS syntax used in the analyses.
• Lines 268-276: while in the previous sections it was reported if ‘channel’ had an effect, this isn’t clear in this section.
More specific methodological comments:
Besides the issue about power analysis above, some other aspects need to be addressed:
• I’m aware the authors used SPSS for the analyses. SPSS is fine and it’s a common tool for statistical analyses. However, it’s limited. For example, I assume the authors estimated mean micro-voltage per participant per condition (as it can be seen in the supplementary file). Would the results hold if a robust estimator, e.g. the median, were used?
• The authors used traditional ANOVAs for RT, ERP and error rates. However, it’s known RTs tend to have a non-normal shape (see 1) so parametric ANOVAs become biased in their estimations. Microvolts also have such shapes. So, such dependent measures call for robust techniques or non-parametric methods (see 2). Would the results hold if these analyses are done?
• Error/accuracy rates are bounded between 0 and 1 so they are akin to a Beta distribution; not a normal distribution. ANOVAS assume a normal distribution; so then again, biased results are obtained. The ideal model is a Beta regression. Would the results hold if a Beta regression were applied? (see ‘betareg’ package in R)
If the authors are in capacity to bring in the expertise of a statistician who uses R, these analyses could be easily done. In case this isn’t possible, the authors should explicitly acknowledge these limitations and warnings.
I recommend future studies include what’s called ‘multiverse analysis’ (see 3); that is, at least 3 statistical analyses should be performed on the data in order to come up with a pattern of results. I recommend a brief mention in the discussion section to this and other methodological matters stated herein.

References
1. Marmolejo-Ramos, F., Cousineau, D., Benites, L., & Maehara, R. (2015). On the efficacy of procedures to normalise Ex-Gaussian distributions. Frontiers in Psychology, 5 (1548): DOI: 10.3389/fpsyg.2014.01548
2. Mair, P., Wilcox, R. Robust statistical methods in R using the WRS2 package. Behav Res 52, 464–488 (2020). https://doi.org/10.3758/s13428-019-01246-w
3. Steegen, S., Tuerlinckx, F., Gelman, A., & Vanpaemel, W. (2016). Increasing Transparency Through a Multiverse Analysis. Perspectives on Psychological Science, 11(5), 702–712. https://doi.org/10.1177/1745691616658637

Reviewer 2 ·

Basic reporting

- The manuscript requires a comprehensive revision in English, as some sentences do not make sense grammatically and especially in the discussion some of the sentence structure does not make sense
- Figure 1 requires more information in the caption than just "stimuli". I would also recommend a legend describing what can be seen on the stimuli and which of them are congruent and incongruent.
- On page 6, beginning with line 84 "Conversely,..." to line 89 requires references.
- Page 6, line 92 "therefore, low-power...". This information is central to the question. I would consider more information on this topic. Last sentence of this paragraph: cannot see the association to the paragraph. Please elaborate this
- Page 6, line 94 “The previous studies..”. Please cite these studies.
- As for the emotional Stroop task, I would add some background information, i.e. what its purpose is classically (this can also be in the methods section and does not have to be in the introduction). In particular, give a better overview of the behavioral interference caused by the semantic incompatibility and that the habitual/automatic reaction (reading) should be inhibited in favor of slower voluntary reactions (emotion recognition). I also recommend citing further literature that has often used the emotional Stroop (such as: Stickel et al., 2019 doi:10.1016/J.BIOPSYCHO.2019.02.008; Chechko et al. 2012 doi:10.1371/journal.pone.0038155, Chechko et al., 2013 doi:10.1016/j.jad.2013.01.013)
- Discussion: The explanation why power has no effect on selective attention refers to the task "reading the word". However, the Stroop task is to ignore the word and recognize the emotion. Therefore I do not understand this explanation. It would also be more helpful to know if the different power conditions react differently to the emotions.
- The discussion of the results refers repeatedly to literature on emotion recognition. Here I wonder again why the authors decided to ignore the different emotions in the task in the analyses.
- I'm not quite sure what is meant by the limitations regarding the power of the targets and the power control condition. This should be described in more detail.

Experimental design

- Please specify in the Stroop task which of the ISIs contained a fixation cross and which contained a blank screen.
- Were there direct repetitions of the same face-word combinations, given this could lead to priming effects?
- I would add more information about power manipulation. Were the participants instructed which experiences count here and how long back in the past. Did only positive experiences count? Was this instructed in a standardized way?

Validity of the findings

- Information on statistical analyses for the behavioral analysis is completely missing. Which programs were used? What calculations were made? It is not enough to simply write in the results that an ANOVA was calculated, because basic information about the calculation process is missing.
- In the abstract it is mentioned that the emotional valence of the faces was not considered. But I wonder why? Is there any reason why the reaction times for the different emotions were not calculated and why no interactions with the social power and EEG components were calculated? Numerous studies (eg. Stickel et al., 2019, Chechko et al., 2013, 2013) describe, for example, that the emotion fear, regardless of whether it is presented congruently or incongruently, has longer reaction times than happy.
- I cannot find the results for the main effect of power and interaction with the interferences.

---

## Round 0.2 · Minor Revisions

I think what is being requested by Reviewer is a very minor revision. Please find below.

·

Basic reporting

although the authors state the manuscript has been proofread, i spotted the institutional affiliation of author 1 is spelled incorrectly... it could be the case there are other typos in the document

Experimental design

seems fine

Validity of the findings

seems fine

Additional comments

comments here

= ensure the english level is acceptable
= the authors state 'non-parametric tests' were used'; please specify the name of those tests
= state where appropriate that data were checked for normality
= along with the SPSS data files (already provided), authors should upload the SPSS's output files. Such files also show the SPSS syntax. This way anyone can easily reproduce the analyses in SPSS

---

## Round 0.3 · accepted · Accept

I would like to express my deepest gratitude to you for submitting your interesting work to PeerJ.